# Industrial Structural Change, the Urban-Rural Income Gap, and Its Regional Heterogeneity: Based on Panel Data from Thirteen Prefecture-Level Cities in Jiangsu Province, 2006-2019

**Tong Jin**
University of Wisconsin-Madison
tong.jin@wisc.edu

## Abstract

With China's economy shifting from high-speed development to high-quality development, balanced urban and rural progress has become crucial in economic development. This paper analyzes the impact of the three major industries on the urban-rural income gap using panel data from thirteen prefecture-level cities in Jiangsu Province from 2006 to 2019, employing an individual fixed effects model. The urban-rural income gap is measured using the Thiel index, which quantitatively captures income distribution disparities by considering both income and population structures. From the provincial perspective, the development of primary and secondary industries effectively suppresses the widening of the urban-rural income gap. In contrast, due to regional heterogeneity in economic development and industrial structures, the impact varies between southern and northern Jiangsu. These findings suggest that targeted industrial policies considering regional differences could play a significant role in narrowing the urban-rural income gap, informing provincial and national strategies for equitable development.

## I. Introduction

If China's economic growth is a success story in the history of human economic development, then China's income distribution is an unsuccessful one. According to data from the website of the National Bureau of Statistics, the ratio of per capita disposable income of urban residents to per capita net income of rural residents in China was 2.56:1 in 1978; in 2008, this figure had come to 3.11:1, and although this figure has fallen in recent years, it still remains at a relatively high level. There are numerous reasons why the urban-rural income gap in China has continued to deteriorate after the reform and opening up. At its root, the urban-rural income gap stems from the urban-rural differences in returns to capital, labour, and other factors. In the context of urbanisation, the gradual consolidation of the urban-rural dichotomy has led to a significant skewing of resource allocation and industrial deployment between the urban and rural sectors, resulting in a massive labour drain from rural areas and stagnation of industrial development. Some researchers have found that the development of the three major industries in the national economy and industrial restructuring affect the distribution of income between urban and rural areas by changing the employment structure of the population. At the same time, an excessive urban-rural income gap can also have a dampening effect on socio-economic development. Therefore, in the process of China's economic development from high speed to high quality development, the impact of the development of the three major industries and industrial restructuring on the urban-rural income gap has become a topic of high importance in regional economics and the relevant studies by researchers have yielded theoretical results that cannot be ignored. Comparing Jiangsu Province with other regions in China, similar patterns of urban-rural income disparity can be observed, but the extent and drivers of this gap vary significantly. For instance, provinces like Guangdong and Zhejiang have also experienced rapid industrialization, but with different industrial structures and policy approaches. By focusing on Jiangsu Province, this study aims to understand whether the patterns observed are unique or part of broader national trends, providing insights that could be applicable to other regions facing similar challenges.

## II. Literature Review

After the reform and opening up, with the advancement of science and technology and the improvement of social productivity, the three major industries have developed at a rapid pace, which has had a certain degree of impact on the income gap between urban and rural areas. (Xue, 1997) studied the relationship between the urban-rural income gap and economic growth. By analysing relevant economic data from 1978-1997, he found that the widening of the urban-rural income gap was related to factors such as low prices of agricultural products and the rapid rise in the wages of urban workers, and found that there was a negative effect on long-term economic growth when the urban-rural income gap was too large. Based on panel data on the income gap between urban and rural residents, (Ren and Wang, 2014) argue that the optimisation of industrial structure is conducive to reducing the income gap between urban and rural residents. (Su, 2002) argues that industrial structure

upgrading is not conducive to reducing the urban-rural income gap in the short run, but in the long run, industrial structure upgrading is a requirement for economic development and is conducive to reducing the urban-rural income gap.

The interaction between industrial transformation and upgrading, labour mobility between different industries, and the urban-rural income gap. (Browning and Singelmann, 1978) point out that changes in the industrial structure of the national economy are closely related to changes in the structure of employment, and that industry (the mainstay of the secondary sector) and services (the mainstay of the tertiary sector) in the urban economy are the basis for structural changes in the economy, and that the "service-oriented transformation" of the employment structure directly leads to the structural transformation of the national economy. However, some scholars hold the opposite view. (Zhao et al., 2018) empirically analysed the impact of industrial structure upgrading on the urban-rural income gap in China through inter-provincial panel data in China, and the results showed that there was a significant negative impact. (Yang et al., 2018) also finds that as the industrial structure continues to upgrade, higher prices for factors of production such as land and labour in cities, prompting industries to move to lower-cost rural areas, will increase employment opportunities and pay levels for rural residents, thereby curbing the widening of the urban-rural income gap.

The above studies have fruitfully explored the interplay between structural transformation, development, and the urban-rural income gap in China's three major industries, and this paper will expand on existing research in the following areas:

First, while most of the existing literature studies the impact on the urban-rural income gap from a single perspective of industrial structure upgrading or economic development, this paper chooses to introduce variables such as economic growth, urbanisation rate, social insurance expenditure, agriculture, forestry and water affairs expenditure, and industrial structure development in combination to explore the impact on the urban-rural income gap under the combined effect.

Second, existing studies have mainly focused on overall national economic growth using inter-provincial panel data to empirically analyse the urban-rural income gap, and fewer have used prefecture-level city data to create panel datasets for analysis. Therefore, after referring to the economic base and industrial development level of each province, we selected 13 prefecture-level cities in Jiangsu Province to build a panel data model for empirical analysis. In addition, based on the heterogeneity of the economic development of each region in Jiangsu Province, we divide the region into two major regions, namely Southern Jiangsu and Northern Jiangsu, so as to explain the mechanism of the impact of the development of the three major industries on the urban-rural income gap

in each region in the hope of providing theoretical support for narrowing the urban-rural income gap.

## III. Indicator Selection and Modelling

### A. Selection of Indicators

In order to explore the role of industrial structural transformation and the development of the three major industries on the urban-rural income gap, the following indicators were selected:

**Explained variable**: The urban-rural income gap (IG) was chosen as the explanatory variable, measured by the Thiel index based on urban-rural population and urban-rural income, combined with income structure and population structure, and calculated as follows:

$$IG = i = \frac{1}{2}\left(\frac{Y_i}{Y}\right)\ln\left(\frac{Y_i L_i}{YL}\right)$$

Where $Y_1 L_1$ denotes the total income of urban residents and the total urban population, $Y_2 L_2$ denotes the total income of rural residents and the total rural population, and $YL$ denotes the total income of residents and the total population. The higher the IG value, the greater the gap between urban and rural incomes.

**Explanatory variable**: The value added of the primary, secondary, and tertiary industries are chosen as explanatory variables ($X_1$, $X_2$, $X_3$), and to attenuate the effects of abnormal fluctuations in the time series, all three variables are taken in logarithmic form ($\ln X_1$, $\ln X_2$, $\ln X_3$).

**Control variable**: In this paper, after referring to existing studies, I decided to choose the logarithmic form of general public service expenditure ($\ln PS$), the logarithmic form of social insurance and employment expenditure ($\ln SI$), the logarithmic form of agriculture, forestry, and water affairs expenditure ($\ln A$), and the urbanisation rate (URL) as control variables.

### B. Data Sources and Descriptive Statistics

The data is sourced from the Jiangsu Provincial Bureau of Statistics, prefecture-level city bureaus, and the Wind database, including panel data for 13 prefecture-level cities in Jiangsu Province with a small amount of missing data calculated using year-on-year growth data from the following year to fill in the data, with basic descriptive statistics for each variable shown in Table I.

### C. Model Building

Existing research shows that the urban-rural income gap is not only influenced by the value added of the three major industries, but also closely related to general public service expenditure, social security and employment expenditure, agriculture, forestry and water affairs expenditure, and urbanisation rate. In order to focus on the influence of industrial structure changes on the urban-rural income gap, this paper takes the urban-rural income gap as the explanatory variable, the value added of the three major industries

| VARIABLES | N | MEAN | SD | MIN | MAX |
|---|---|---|---|---|---|
| IG | 182 | 0.0535 | 0.0215 | 0.0258 | 0.119 |
| $\ln X_1$ | 182 | 5.288 | 0.556 | 3.694 | 6.526 |
| $\ln X_2$ | 182 | 7.426 | 0.764 | 5.354 | 9.119 |
| $\ln X_3$ | 182 | 7.277 | 0.887 | 5.053 | 9.201 |
| $\ln PS$ | 182 | 3.708 | 0.703 | 1.823 | 5.306 |
| $\ln SI$ | 182 | 3.538 | 0.912 | 0.307 | 5.281 |
| $\ln A$ | 182 | 3.564 | 0.734 | 1.208 | 4.822 |
| URL | 182 | 0.613 | 0.109 | 0.324 | 0.832 |

TABLE I: Descriptive statistics

| Variable Conclusion | HT | IPS | LLC |
|---|---|---|---|
| IG Unstable | 0.5671 | 0.5103 | -4.3580*** |
| $\ln X_1$ Stable | 1.6903 | -4.4206*** | -10.2966*** |
| $\ln X_2$ Stable | 1.8882 | -7.3112*** | -9.2212*** |
| $\ln X_3$ Stable | 2.3942 | -6.2041*** | -10.5160*** |
| $\ln PS$ Unstable | 1.4122 | 1.0698 | -1.5118 |
| $\ln SI$ Stable | 2.0298 | -1.3624* | -5.0284*** |
| $\ln A$ Stable | 1.212 | -2.7356*** | -4.7041*** |
| URL Unstable | 2.5639 | 2.3822 | -3.9636*** |

TABLE II: Test of Stationarity

as the explanatory variable, and the rest of the variables as control variables, and establishes the following panel model:

$$\text{IG}_{it} = \alpha_0 + \alpha_i + \beta_1 \ln X_{1it} + \beta_2 \ln X_{2it} + \beta_3 \ln X_{3it} + \sum_{k=1}^{K} \gamma_k X_{kit} + \mu_{it}$$

Where $i$ denotes different prefecture-level cities, $t$ denotes year, $IG_{it}$ denotes Thiel index of the explained variable urban-rural income gap, $\ln X_1 it$, $\ln X_2 it$, $\ln X_3 it$ denotes the logarithmic forms of the explanatory variables primary, secondary, and tertiary value added respectively, $\beta_1$, $\beta_2$, $\beta_3$ are their corresponding coefficients, $X_k it$ is the matrix of control variables, $k$ represents the number of control variables, $\gamma_k$ denotes coefficient of each control variables, $\alpha_0$ is the intercept term, $\alpha_i$ represents the individual effect, $\mu_{it}$ is the random error term.

## IV. Empirical Analysis

### A. Analysis of Full Sample Empirical Results

For panel data, unstable variables may lead to pseudo-regression phenomena, making the estimation results inaccurate, so a test of stationarity is performed first. According to the characteristics of the sample data, I decided to use HT test, LLC test, and IPS test to conduct their stationarity. Table II shows the results.

The voting method was used to determine whether the variables were stable or not, and it was found that only three variables (IG, $\ln PS$, URL) were not stable, while the rest of the variables were stable at 10% and above significance level. By further testing, the unstable variables were all found to be I(1) and the model could pass the Kao test of cointegration and Westerlund test of cointegration.

The heterogeneity of the municipalities in the model results in different intercept terms, and the model can be classified as a fixed-effects model or a random-effects model depending on the characteristics of the intercept terms. The Hausman test was then performed and the results indicated that the original hypothesis

of the model being a random effect should be rejected at a significance level of 0.1, so a variable intercept model with individual fixed effects should be built for the analysis.

The full sample regression results are shown in Table III, where regression equation (1) does not introduce control variables and regression equation (2) is the result of adding control variables.

| | (1) | (2) |
|---|---|---|
| $\ln X_1$ | -0.0234** | -0.0442*** |
| | (0.0097) | (0.0096) |
| $\ln X_2$ | -0.0263** | 0.0150 |
| | (0.0116) | (0.0133) |
| $\ln X_3$ | 0.0072 | 0.0209** |
| | (0.0097) | (0.0093) |
| $\ln PS$ | | -0.0114*** |
| | | (0.0032) |
| $\ln SI$ | | -0.0002 |
| | | (0.0039) |
| $\ln A$ | | 0.0004 |
| | | (0.0031) |
| URL | | -0.2302*** |
| | | (0.0432) |
| _cons | 0.3209*** | 0.2064*** |
| | (0.0293) | (0.0419) |
| N | 182 | 182 |
| $R^2$ | 0.709 | 0.764 |

TABLE III: Full sample regression analysis

Note: *p<0.1, **p<0.05, ***p<0.01. Values in parentheses correspond to t-statistics. _cons is the intercept term.

From regression (1), it can be seen that the development of the primary and secondary industries has a significant effect on reducing the urban-rural income gap. For every 1% increase in the value added of the primary sector, the urban-rural income gap narrowed by an average of 0.000234 units, indicating that the primary sector, which is dominated by agriculture, can to a certain extent weaken the degree of deviation in employment structure and industrial structure between urban and rural areas. At the same time, for every 1% increase in the value added of the secondary sector,

the urban-rural income gap narrowed by an average of 0.000263 units, indicating that the development of the secondary sector, mainly industry, has also improved the degree of deviation in the industrial structure between urban and rural areas to some extent. In contrast, the effect of the tertiary sector on the urban-rural income gap was not significant. As one of the most economically developed provinces in China, Jiangsu is very well developed in agriculture and is located in the lower reaches of the Yangtze and Huai rivers, mostly in the plains, which is suitable for mechanised farming. With the continuous improvement of agricultural production efficiency, the modernisation of agriculture and rural areas has been improved, thus curbing the further widening of the income gap between urban and rural residents. Since the reform and opening up, Jiangsu has gradually formed an industrial structure with the secondary industry as the core, and labour-intensive industries have absorbed a large amount of rural labour, thus curbing the further widening of the urban-rural income gap.

After adding the control variables of social security and employment expenditure, general public service expenditure, and expenditure on agriculture, forestry, and water affairs, the effect of the primary sector on the urban-rural income gap increases significantly, while the effect of the secondary sector on the urban-rural income gap shifts from significantly smaller to insignificant, and the sign of the regression coefficient of the tertiary sector on the urban-rural income gap changes from insignificant to significantly larger. The phenomenon shows that the development of the primary industry in Jiangsu Province in the process of implementing the rural revitalisation strategy and promoting the transformation and upgrading of the primary sector has directly brought about the expansion of the scale of agricultural production and the improvement of agricultural production efficiency, which is inseparable from the natural environmental conditions of Jiangsu Province and thus reflected in its direct effect of raising the income of rural residents. The widening effect of the tertiary sector on the urban-rural income gap stems from the high demand for labour in the service sector (the core of the tertiary sector), which tends to absorb more non-rural labour, while rural labourers are limited by their lower literacy level and mostly remain in low-productivity and low value-added industries in the primary and secondary sectors, leading to a further widening of the income gap between urban and rural residents. Both control variables $\ln PS$ and URL show a significant reduction after the inclusion of the control variables. As a necessary expenditure to ensure equity for all citizens, general public services have been significantly effective in reducing the urban-rural income gap, with each 1% increase in general public service expenditure reducing the urban-rural income gap by an average of 0.0114 units. In the process of urbanisation, the influx of rural people into cities can effectively improve the income of rural residents, thus

reducing the urban-rural income gap. At the Jiangsu province level, the development of the primary and tertiary industries, $\ln PS$ and URL, have a significant effect on the urban-rural income gap, and the control variables except $\ln PS$ and URL all have an indirect effect on the urban-rural income gap by influencing the development of the three major industries.

## B. Analysis of Empirical Results for Sub-Regional Samples

Considering the uneven distribution of the three major industries within Jiangsu Province and the possible regional differences, Jiangsu was divided into two regions, South Jiangsu and North Jiangsu, for empirical analysis and the results were obtained as shown in Table IV.

|  | South Jiangsu | North Jiangsu |
|---|---|---|
| $\ln X_1$ | -0.0122** | -0.0670** |
|  | (0.0051) | (0.0207) |
| $\ln X_2$ | 0.0051 | 0.0245 |
|  | (0.0088) | (0.0225) |
| $\ln X_3$ | 0.0073 | 0.0067 |
|  | (0.0049) | (0.0217) |
| $\ln PS$ | -0.0052** | -0.0153** |
|  | (0.0023) | (0.0060) |
| $\ln SI$ | -0.0062*** | -0.0035 |
|  | (0.0017) | (0.0092) |
| $\ln A$ | 0.0047** | -0.0037 |
|  | (0.0020) | (0.0053) |
| URL | -0.1731*** | -0.0019 |
|  | (0.0290) | (0.0978) |
| _cons | 0.1567*** | 0.2946*** |
|  | (0.0301) | (0.0766) |
| N | 84 | 98 |
| $R^2$ | 0.938 | 0.784 |

TABLE IV: Regional sample regression

Note: *p<0.1, **p<0.05, ***p<0.01. Values in parentheses correspond to t-statistics. _cons is the intercept term.

From the regression results of the regional sample, the development of primary industries in southern Jiangsu has a significant effect on reducing the urban-rural income gap, while the effect of secondary and tertiary industries on the urban-rural income gap is not significant; the development of primary industries in northern Jiangsu also has a significant effect on reducing the urban-rural income gap and the suppression effect is more obvious than that in southern Jiangsu, while the effect of secondary and tertiary industries on the urban-rural income gap is not significant.

Due to the difference in the degree of economic development and the slightly different focus of industries in the north and south of Jiangsu Province, the degree of suppression of the urban-rural income gap by the development of primary industries in the south and north of Jiangsu Province differs greatly, while the suppression of the urban-rural income gap by the development of primary industries in the south of Jiangsu Province, where manufacturing is the core of economic construction, is relatively smaller. The

main reason is that the industrial structure of southern Jiangsu is highly concentrated in the secondary industry, with manufacturing as the core, and its industrial development is among the most advanced in China. On the contrary, due to geographical factors such as well-developed water systems, the degree of modernisation of agriculture is lower, and the production efficiency is less enhanced, thus failing to realise the feed-back effect of industry on agriculture and therefore has less ability to suppress the urban-rural income gap. Therefore, it is important for southern Jiangsu to pay more attention to the rationalisation of the industrial structure, to actively promote the transformation and upgrading of the primary industry, and to take advantage of its own complete manufacturing system to realise the feed-back effect of industry on agriculture. At the same time, the primary industry in northern Jiangsu accounts for a relatively high proportion of the region's national economy. Under the thrust of agricultural supply-side reform, the rate of transformation and upgrading of the primary industry exceeds that of the secondary and tertiary industries, so northern Jiangsu should centre its industrial structure transformation on the secondary and tertiary industries, accelerate the process of industrialisation and modernisation of the service system, adjust the dual employment structure between urban and rural areas, and continue to promote the rationalisation of the industrial structure, so as to further narrow the gap between urban and rural incomes.

In addition to the significant impact of the development of the three major industries on the urban-rural income gap, the control variables of each region also have a certain degree of impact on the urban-rural income gap. General public affairs expenditure shows a suppressive effect on the urban-rural income gap in southern Jiangsu as well as in northern Jiangsu, which is consistent with the results in the full-sample regression, indicating that the general public affairs expenditure of the Jiangsu government can guarantee citizens' enjoyment of equal rights. Social insurance and employment expenditure can at least protect the income of rural residents and fundamentally reduce the urban-rural income gap, but it is influenced by other economic factors of the region and there are significant regional differences between southern and northern Jiangsu and its effect on reducing the urban-rural income gap is significant in southern Jiangsu but not in northern Jiangsu. The expenditure on agriculture, forestry and water affairs is manifested in the southern part of Jiangsu Province to promote further widening of the income gap between urban and rural areas. This is probably due to the fact that the existing input structure in southern Jiangsu is not reasonable enough, with a large amount of funds being used for emergency and disaster prevention projects such as flood control and drought relief, and insufficient investment in farmland infrastructure that directly serves agricultural production, thus indirectly leading to an increasingly unbalanced allocation of resources between urban and rural areas. In contrast, this control variable did not perform significantly in northern Jiangsu. The urbanisation rate significantly widens the urban-rural income gap in both southern and northern Jiangsu, with the widening effect being stronger in northern Jiangsu than in southern Jiangsu. As the level of urbanisation in Jiangsu Province gradually increases from north to south, the urbanisation process in northern Jiangsu Province is relatively slow, and its industrial structure is dominated by heavy industries with higher value-added, with a higher proportion of workers entering the city to work in low value-added jobs, and the employment structure of urban and rural residents has not essentially changed, which further widens the urban-rural income gap. Southern Jiangsu, on the other hand, has relatively low barriers to employment in the industry due to its core structural orientation of light industry, and relatively weak ability to widen the urban-rural income gap.

The regression results reveal that the primary industry's development significantly reduces the urban-rural income gap in both southern and northern Jiangsu, but the effect is more pronounced in the northern region. This difference can be attributed to the varying economic structures and levels of industrialization between the two regions. Northern Jiangsu relies more heavily on agriculture, and improvements in agricultural productivity directly enhance rural incomes, thereby reducing the income gap more substantially than in the industrialized south.

The mechanism by which public service expenditure contributes to reducing the income gap lies in its role in providing essential services such as education, healthcare, and social security. By allocating more resources to public services, the government can improve living standards in rural areas, enhance human capital, and provide safety nets that disproportionately benefit lower-income rural residents. This targeted expenditure helps to level the playing field between urban and rural populations.

## V. Conclusion and Suggestion

This paper analyses the impact of the development of the three major industries on the urban-rural income gap by constructing a panel data model with individual fixed effects using data from 2006-2019 for a sample of Jiangsu Province and a sub-regional sample, controlling for the impact of social security and employment expenditure, general public service expenditure, expenditure on agriculture, forestry and water affairs, and urbanization rate respectively, with the following main conclusions and suggestions for countermeasures:

There are significant differences in the degree of economic development and industrial structure tendencies within the province of Jiangsu, and each prefecture-level government should designate appro-

priate industrial transformation policies according to the actual local situation. In southern Jiangsu, the industrial structure is highly conversant with rapid development of secondary and tertiary industries, but limited investment in the development of the primary industry, which should be given more attention to make the industrial structure more reasonable and effectively improve the quality of life of rural residents. In contrast, the tertiary industry in northern Jiangsu is poorly developed and rural residents still have plenty of opportunities to engage in high value-added jobs. The social security mechanism for process workers should be further improved, rural residents should be encouraged to study continuously and improve their work efficiency, while the rapid development rate of the primary industry should be maintained and the rural modernisation policy should be further implemented to improve the income of rural residents.

Restructuring the primary sector and improving the productivity of the primary sector has a good inhibiting effect on reducing the income gap between urban and rural areas. According to the different natural conditions of each region, the layout of agricultural production should be adjusted and optimised according to local conditions, and advantageous agriculture with high added value should be vigorously developed to form agricultural industry clusters with agglomeration effect. At the same time, the policy of increasing direct subsidies for farmers and indirect subsidies that help to raise farmers' income levels in agriculture, forestry and water affairs expenditures should be enhanced to ensure stable prices for agricultural products, effectively safeguard farmers' interests, and ensure that the development of the primary industry continues to contribute to the narrowing of the urban-rural income gap.

The existing pattern of development of the tertiary industry in Jiangsu Province, which is oriented towards high-end service industries, has contributed to the widening of the income gap between urban and rural areas, while the balanced distribution of resources in the service industry can help narrow the income gap between urban and rural areas. Vigorously develop rural enterprises and guide them to accelerate technological progress and industrial upgrading. Vigorously develop labour-intensive industries and rural service industries to enhance the ability to absorb rural labour for local and nearby employment. Actively clean up and abolish various discriminatory regulations and unreasonable restrictions on the employment of rural workers, gradually establish a unified urban and rural labour market and a fair and competitive employment system, guide the transfer of rural labour to employment, and increase the labour income of farmers. At the same time, we are increasing investment in public services such as education and healthcare in rural areas, breaking down the barriers of the inherent urban-rural dual structure and effectively improving the quality of life of rural residents.

The dual economic system of urban and rural areas is a necessary path for developing countries in their economic development. The relatively low level of infrastructure development and living standards in rural areas has led to a massive exodus of labour, making rural labour shortages and economic development difficult, further leading to a widening income gap between urban and rural areas, which is a vicious circle. In order to break this cycle, the government needs to take active action to eliminate the disparity between urban and rural residents in terms of employment, education, and healthcare and to blur the border between rural and urban areas in order to move into a modern economic structure at an early date, allowing Jiangsu's economy to develop in a higher quality and quantity.

The findings underscore the importance of formulating region-specific industrial policies to effectively narrow the urban-rural income gap. In southern Jiangsu, policies should focus on integrating advanced agricultural technologies and promoting high-value-added agriculture to enhance rural incomes. In northern Jiangsu, accelerating industrialization and developing the tertiary sector can create more employment opportunities and stimulate income growth for rural residents.

Moreover, increasing public service expenditure in rural areas can address disparities in education, healthcare, and social welfare, providing a more comprehensive approach to reducing income inequality. Policymakers at both provincial and national levels should consider these regional nuances when designing strategies to promote balanced economic development and social equity.

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
