# OpenReview forum: "Industrial Structural Change, the Urban-Rural Income Gap, and Its Regional Heterogeneity: Based on Panel Data from Thirteen Prefecture-Level Cities in Jiangsu Province, 2006-2019"
_IEEE.org/ICIST/2024/Conference — IEEE ICIST 2024 Conference Submission_

### Official Review · Reviewer_gYht · 2024-08-21
**Review Results**

**Rating:** 5
**Confidence:** 4

**Review:**

This paper analyzes the impact of the three major industries on the urban-rural income gap using panel data from 13 prefecture-level cities in Jiangsu Province from 2006 to 2019.

1.What accounts for the difference in the impact of the primary industry on the urban-rural income gap between the southern and northern regions of Jiangsu?

2.What is the mechanism by which public service expenditure significantly contributes to reducing the income gap?

3.How can industry policies be formulated based on regional differences to more effectively narrow the urban-rural income gap?

---

### Official Review · Reviewer_xJ6a · 2024-08-22
**Industrial Structural Change, the Urban-Rural Income Gap, and Its Regional Heterogeneity: Based on Panel Data from Thirteen Prefecture-Level Cities in Jiangsu Province, 2006-2019**

**Rating:** 5
**Confidence:** 4

**Review:**

The article investigates the impact of industrial structural changes on the urban-rural income gap in Jiangsu Province, China, using panel data from 13 prefecture-level cities between 2006 and 2019. The study examines how the development of the primary, secondary, and tertiary industries influences the income disparity between urban and rural areas. The analysis is conducted both at the provincial level and by dividing Jiangsu into southern and northern regions to account for regional heterogeneity. The results indicate that the primary and secondary industries have a significant role in reducing the income gap, particularly in the less developed northern regions. In contrast, the tertiary industry tends to widen the gap, especially in southern Jiangsu, where high-end service industries dominate. The study suggests that targeted policies addressing regional differences in industrial development could help narrow the urban-rural income gap.

Comments:

1.The paper presents a comprehensive analysis, but the presentation of key findings could be made clearer, especially regarding how the results differ between southern and northern Jiangsu. A more concise summary of the regional differences in the conclusion would improve the clarity of the paper.

2.While the methodology is sound, the paper could benefit from a more detailed explanation of the statistical tests used, such as the stationarity tests and cointegration tests, to help readers understand the robustness of the results.

3.The paper effectively reviews existing literature but could further integrate these references into the discussion of results to highlight how the findings contribute to or challenge existing theories on the urban-rural income gap and industrial development.

4.The language is generally clear, but certain sections could be made more concise. Simplifying some of the technical terms and reducing repetition would make the paper more accessible to a broader audience.

---

### Official Review · Reviewer_A69H · 2024-08-22
**Interesting**

**Rating:** 6
**Confidence:** 3

**Review:**

The paper provides valuable insights into the impact of industrial structural changes on the urban-rural income gap in Jiangsu Province, with a strong emphasis on regional heterogeneity. This topic is interesting, the following comments need to further consider: 1. The abstract does not provide detailed definitions of the indicators used, particularly the Thiel index, which is mentioned as a measure of the urban-rural income gap. More information on how this index is calculated and why it was chosen over other potential measures would be helpful. 2. While the study is focused on Jiangsu Province, it would benefit from a brief comparative analysis with other regions in China. This could help contextualize the findings and highlight whether the patterns observed in Jiangsu are unique or part of broader national trends. 3. The abstract does not discuss the potential policy implications of the findings. Given the focus on income inequality and regional development, a brief mention of how these findings could inform policy decisions at the provincial or national level would enhance the practical relevance of the research. 4. The study’s focus on industrial structural changes as the primary driver of the urban-rural income gap may overlook other significant factors, such as education, healthcare, and infrastructure disparities. A more balanced discussion that includes these factors could provide a more comprehensive understanding of the issue.

---

### Decision · Program_Chairs · 2024-09-08

Accept (Oral)